# Comprehensive short and long read sequencing analysis for the Gaucher and Parkinson's disease-associated *GBA* gene

Marco Toffoli [1,10], Xiao Chen[2,9,10], Fritz J. Sedlazeck [3], Chiao-Yin Lee [1], Stephen Mullin[1,4], Abigail Higgins [1], Sofia Koletsi[1], Monica Emili Garcia-Segura[1], Esther Sammler[5,6], Sonja W. Scholz[7,8], Anthony H. V. Schapira[1], Michael A. Eberle [2,9,10✉] & Christos Proukakis [1,10✉]

*GBA* variants carriers are at increased risk of Parkinson's disease (PD) and Lewy body dementia (LBD). The presence of pseudogene *GBAP1* predisposes to structural variants, complicating genetic analysis. We present two methods to resolve recombinant alleles and other variants in *GBA*: Gauchian, a tool for short-read, whole-genome sequencing data analysis, and Oxford Nanopore sequencing after PCR enrichment. Both methods were concordant for 42 samples carrying a range of recombinants and *GBAP1*-related mutations, and Gauchian outperformed the GATK Best Practices pipeline. Applying Gauchian to sequencing of over 10,000 individuals shows that copy number variants (CNVs) spanning *GBAP1* are relatively common in Africans. CNV frequencies in PD and LBD are similar to controls. Gains may coexist with other mutations in patients, and a modifying effect cannot be excluded. Gauchian detects more *GBA* variants in LBD than PD, especially severe ones. These findings highlight the importance of accurate *GBA* analysis in these patients.

[1] Department of Clinical and Movement Neurosciences, Queen Square Institute of Neurology, University College London, London NW3 2PF, United Kingdom. [2] Illumina Inc., San Diego, CA, USA. [3] Human Genome Sequencing Center, Baylor College of Medicine, Houston, TX, USA. [4] Institute of Translational and Stratified Medicine, University of Plymouth School of Medicine, Plymouth, United Kingdom. [5] MRC Protein Phosphorylation and Ubiquitylation Unit, School of Life Sciences, University of Dundee, Dundee, United Kingdom. [6] Molecular and Clinical Medicine, School of Medicine, University of Dundee, Dundee, United Kingdom. [7] Neurodegenerative Diseases Research Unit, National Institute of Neurological Disorders and Stroke, Bethesda, MD 20892, USA. [8] Department of Neurology, Johns Hopkins University Medical Center, Baltimore, MD 21287, USA. [9] Present address: Pacific Biosciences, 1305 O'Brien Dr., Menlo Park, CA 94025, USA. [10] These authors contributed equally: Marco Toffoli, Xiao Chen, Michael A. Eberle, Christos Proukakis. ✉email: meberle@pacbio.com; c.proukakis@ucl.ac.uk

The *GBA* gene encodes the lysosomal enzyme glucocerebrosidase, and biallelic mutations in *GBA* cause the autosomal recessive disorder Gaucher disease (GD [MIM: #230800, #230900 and #231000])[1]. Around 500 disease-causing mutations have been reported, mostly missense changes arising from single nucleotide variants (SNVs)[2]. Heterozygous variants in *GBA* [MIM: *606463] are associated with an increased risk of developing Parkinson's disease (PD)[3], the second most common neurodegenerative disease, and the closely related Lewy body dementia (LBD)[4]. Identifying *GBA* mutations is difficult due to a pseudogene (*GBAP1*) located 6.9 kb downstream[5] which has an overall homology of 96% with *GBA*. This rises to 98% in the region from intron 8 to the 3′-UTR, where there are five identical segments >200 bp each[6]. The high homology predisposes to non-allelic homologous recombination between *GBA* and *GBAP1*, leading to a wide range of structural variants (SV)[7]. These can be non-reciprocal, also termed gene conversion, or reciprocal, resulting in copy number variants (CNV). Throughout this paper, we use the term copy number gain (CNG) for reciprocal duplication alleles where a 20.6 kb long region of DNA between the homology segments of *GBA* and *GBAP1* is multiplied, and copy number loss (CNL) for reciprocal fusion alleles where the same region is deleted, creating *GBA-GBAP1* fusions[7] (see Fig. 1). SVs that disrupt the coding sequence by gene conversion or reciprocal recombination are expected to be pathogenic for GD and risk factors for PD. Conversely, SVs not affecting the coding sequence are not pathogenic, although a modifier effect cannot be excluded[7]. These include CNLs outside the coding region, and all CNGs, which consist of a partial duplication of pseudogene sequence merged with a variable part of the gene, often only the 3′ UTR, with the resulting allele still containing a normal copy of the *GBA* coding region (Fig. 1d). The SV variability and population prevalence remain largely unknown. Pathogenic missense changes in the high homology exon 9–11 region such as the common p.L483P (NC_000001.11:g.155235252 A > G), also known as p.L444P) may arise by gene conversion, rather than simple base substitutions, with pseudogene sequence

incorporated into the gene[7]. We refer to variants corresponding to pseudogene bases in this region as *GBAP1*-like. The most common *GBAP1*-like variants introduced by either gene conversion or reciprocal recombination in the exon 9–11 homology region include p.L483P, p.D448H (NC_000001.11:g.155235727 C > G), c.1263del55 (NC_000001.11:g.155235752_155235806del), RecNciI (which comprises three SNVs: p.L483P, p.A495P and p.Val499=), RecTL (RecNciI and p.D448H) and c.1263del +RecTL (RecNciI, p.D448H and c.1263del55).

Current sequencing approaches to characterise *GBA* have major pitfalls, and to date, no single approach has fully resolved recombinants[6]. The correct alignment of short reads when there is a highly similar pseudogene is intrinsically problematic, and *GBA* is challenging in exome and whole-genome sequencing (WGS)[8–10], containing camouflaged regions[11]. Moreover, the reliability of the standard WGS secondary analysis pipelines such as the Genome Analysis Toolkit best practice workflow[12] has not been formally assessed. Targeted short-read sequencing approaches are also possible but may require forced alignment to *GBA* and visual inspection and Sanger validation to detect recombinant variants, and are not likely to provide copy number information[6,13]. We have already performed refinement of the Illumina WGS analysis for other difficult regions due to sequence homology, demonstrating reliable resolution of SVs in such regions on Illumina WGS data in the *SMN1*[MIM: *600354]/*SMN2*[MIM: *601627] genes in spinal muscular atrophy[14] and the pharmacogene *CYP2D6*[MIM: *124030][15]. We also previously reported a method for *GBA* analysis using enrichment by long-range PCR, followed by sequencing on the Oxford Nanopore Technologies (ONT) MinION[16], which reliably detected SNVs, including *GBAP1*-like variants, and could also detect non-reciprocal recombinants, but not reciprocal recombinants (Fig. 1)[16].

To overcome these limitations and improve the characterisation of *GBA* at scale, we have developed refined pipelines based on either targeted analysis of short-read (Illumina) WGS data or targeted long-read (ONT) single-molecule sequencing. For

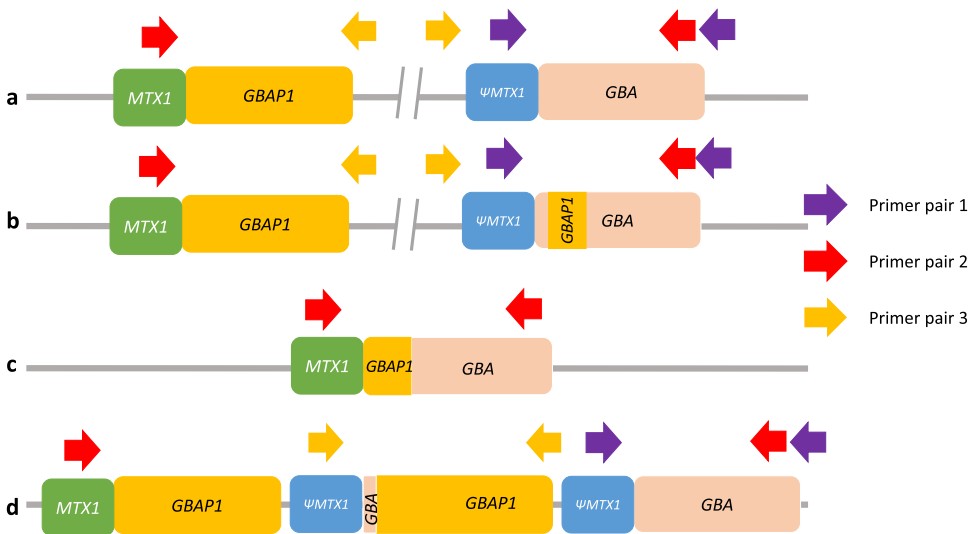

**Fig. 1 Schematic illustration of the different types of *GBA* recombinant alleles and positions of PCR primers used to detect them with ONT.** Not to scale, corresponding roughly to g.chr1:155,210,000-155,245,000. **a** Wild-type allele. Only primer pair 1 will produce an amplicon. **b** Non-reciprocal recombination (gene conversion). Similar to non-recombinant alleles, only primer pair 1 will produce an amplicon. **c** Reciprocal crossover between gene and pseudogene resulting in a 20.6 kb deletion (CNL). Only primer pair 2 will produce an amplicon. **d** Reciprocal crossover between gene and pseudogene resulting in a 20.6 kb duplication (CNG). Both primer pair 1 and primer pair 3 will produce amplicons. Note that the normal allele is present and that amplification with primer pair 3 will produce an amplicon independently of the number of copy number gains. CNG copy number gain, CNL copy number loss.

Illumina data, we present and validate 'Gauchian', a novel algorithm for *GBA* locus analysis which can reliably resolve SVs and *GBAP1*-like variants. For ONT data, we have addressed the problem of reciprocal recombinants by using PCR primers designed to amplify CNGs and CNLs when they exist. We validated these methods and then applied them to large PD, LBD, and population control samples. We demonstrate that complete resolution of all variant types in *GBA* is possible using either Gauchian analysis of Illumina WGS data or targeted ONT sequencing. Finally, we confirm that *GBA* variants are more common in LBD than in PD, we report the frequency of CNVs in different populations and suggest that a possible modifier role of CNG in PD and LBD merits further study. Both methods finally enable precise characterisation of *GBA* at scale, thus driving the identification of causative variants forward.

## Results

**Cross-validation confirms both Gauchian and ONT methods.** To select appropriate samples for validation of Gauchian with a broad range of mutations, we first obtained Gauchian results on 1000 Genomes Project (1kGP) samples and the Accelerating Medicines Partnership Parkinson's Disease (AMP-PD) PD and control cohorts. We selected 37 of these for validation by ONT targeted sequencing. These included 15 samples from 1kGP with CNVs or gene conversions, and 22 samples from Parkinson's Progression Markers Initiative (PPMI—included in AMP-PD cohort), where Gauchian showed CNVs or *GBAP1*-related variants ($n = 7$), or was discordant with available Broad Institute's Joint Genotyping pipeline (referred to as BWA-GATK in this paper) results ($n = 4$), or no mutation was reported by Gauchian or BWA-GATK ($n = 11$). Additionally, for 5 brain DNA samples analysed first by ONT with recombinations or *GBAP1*-related mutations, we performed Illumina WGS and Gauchian analysis. All 42 Gauchian results were consistent with ONT. Within these validation samples, Gauchian reported five CNL, which included one in which the p.L483P was also found, one resulting in the pathogenic RecNciI, and 14 CNGs, including two samples which also carried a gene conversion, and one which carried p.L483P. Additionally, in 11 samples Gauchian called *GBAP1*-like variant calls within *GBA*, including two gene conversions. The remaining 12 samples were wild-type calls. Notably, Gauchian and ONT gave concordant results in two samples where previous BWA-GATK analysis had missed p.L483P, and two where BWA-GATK had wrongly called p.A495P (NC_000001.11:155235216:C:G). Results of cross-validation of the two methods are reported in Table 1.

To obtain further orthogonal validation, we used digital PCR (dPCR) for copy number estimation of the 20.6 kb region involved in recombination in six samples with a range of copy numbers. These included four samples where Gauchian and ONT both detected a CNG (additional copy numbers 1, 3, 5 and 6), and two where we only had ONT data, one with a CNL, and one with no CNV. The results were fully concordant (Supplementary Table 1). Finally, we applied ONT sequencing with PCR-free enrichment by adaptive sampling (UNCALLED)[17] to four reciprocal recombinants, two CNG and two CNL (one pathogenic and one non-pathogenic). Inspection of the resulting alignments confirmed the presence of the SV and the breakpoints of CNL alleles (Supplementary Fig. 1).

**Detection of all classes of *GBA* variants with targeted ONT long-read sequencing.** To fully resolve the *GBA* gene with ONT, we used three pairs of primers, as detailed in the methods section. Primer pair 1 is designed to amplify the *GBA* gene, as previously described[16]. Primer pairs 2 and 3 are used to detect CNL and CNG alleles, respectively.

We analysed 397 samples from PD or GD patients, their relatives, and controls (Supplementary Table 2) using all three pairs of PCR primers, followed by ONT amplicon sequencing. These included 95 individuals previously sequenced with ONT using only primer pair 1[16]. All results are shown in Supplementary Table 3. We detected two c.1263del + RecTL alleles and one RecNciI allele arising from gene conversion. Additionally, we also detected 94 coding or splice site SNVs, including the pathogenic mutations p.N409S (NC_000001.11:g.155235843 T > G, also known as p.N370S; 38% of all SNV detected) and the *GBAP1*-like p.L483P (20%), and the PD risk alleles p.E365K (NC_000001.11:g.155236376 C > T, also known as p.E326K, 16%) and p.T408M (NC_000001.11:g.155236246 G > A, also known as p.T369M, 4%). Notably, we also detected c.84dupG (NC_000001.11:g.155240661dup), the most common pathogenic indel in the *GBA* gene. As homopolymer regions are challenging for ONT[18], and *GBA* has two coding poly-G stretches, we devised a method to detect variants within these (Supplementary Fig. 2). This additional analysis identified one single base deletion and one SNV within homopolymers that would have been missed by our old ONT pipeline (c.413delC and p.P68=, NC_000001.11:g.155239661del and NC_000001.11:g.155239989 C > T, Supplementary Fig. 2). We detected CNLs using primer pair 2 in nine samples. According to the position of the breakpoints (Supplementary Fig. 3), five of them were pathogenic, and four were not. Two non-pathogenic CNLs were *in cis* with p.L483P, a pattern already described[7]. We also detected a CNG using primer pair 3 in seven samples, four of which also carried a *GBAP1*-like variant (two c.1263del+RecTL and two p.L483P).

**Comprehensive *GBA* analysis by Gauchian in short-read WGS population data.** A total of 10623 samples were analysed with Gauchian, including 2504 samples from the 1kGP cohort, 2325 PD and 2598 LBD samples from the AMP-PD knowledge portal, and their respective controls. We identified 55 non-pathogenic CNLs and 146 CNGs (roughly correspond to DGV variant accessions dgv55e214 and esv3587619; Table 2). Additionally, we detected 97 *GBAP1*-like variants (including those generated by pathogenic CNL or gene conversion) in the exons 9–11 homology region of *GBA* in all three cohorts (Table 3 and Supplementary Table 4).

BWA-GATK variant calls were available for all AMP-PD samples analysed by Gauchian. For all PD and LBD case/control populations, BWA-GATK called 44 *GBAP1*-like variants, and Gauchian called 86, almost doubling the variant calls. Due to the sequence homology and misalignment of reads in exons 9–11, the BWA-GATK pipeline under-called all *GBAP1*-like variants except p.A495P and p.D448H. For p.A495P, GATK called 11 false positives (including two confirmed as false positives by ONT amplicon sequencing- see earlier) and for p.D448H BWA-GATK called two false positives. The false-positive calls by BWA-GATK are due to alignment errors caused by *GBAP1* haplotypes containing *GBA* bases (see Fig. 2c). Gauchian also detected other coding SNVs and indels that are not *GBAP1*-like in the three cohorts (see Supplementary Table 5 for all variants). All these calls were concordant with BWA-GATK except in one sample where Gauchian called p.L483R, a rare pathogenic variant in the same codon as the common *GBAP1*-like p.L483P, but BWA-GATK did not. This variant is in the exon 9–11 homology region, and the variant reads misaligned to *GBAP1*, causing the false-negative by BWA-GATK (Supplementary Fig. 4).

To investigate the Mendelian consistency of Gauchian calls and the prevalence of *de novo* variants, we analysed 602 trios in

**Table 1 Details of cross-validation between Gauchian and ONT.**

| Sample[a] | CN change | Other variants | CNV | Variant type | Number of samples |
|---|---|---|---|---|---|
| NA20756 | 1 | | Gain | CNG with no other variant | 11 |
| HG01912 | 3 | | | | |
| HG01889 | 5 | | | | |
| HG02284 | 6 | | | | |
| HG03547 | 3 | | | | |
| NA19909 | 4 | | | | |
| HG03895 | 1 | | | | |
| NA18917 | 2 | | | | |
| NA19711 | 2 | | | | |
| HG03575 | 4 | | | | |
| Brain-S1 | 1 | | | | |
| PP-3307* | 1 | p.L483P | | CNG + SNV | 1 |
| Brain-S2 | 4 | c.1263del +RecTL | | CNG + c.1263del+RecTL conversion | 2 |
| Brain-S3 | 4 | c.1263del +RecTL | | | |
| HG03428 | −1 | | Loss | CNL, non-pathogenic | 3 |
| NA19024 | −1 | | | | |
| PP-12224 | −1 | | | | |
| HG00422 | −1 | RecNcil | | Pathogenic CNL (RecNcil CNL) | 1 |
| Brain-S4 | −1 | p.L483P | | Non-pathogenic CNL + p.L483P | 1 |
| HG00119 | 0 | c.1263del +RecTL | No CN change | Gene conversion | 2 |
| HG00115 | 0 | c.1263del +RecTL | | | |
| PP-3420 | 0 | p.L483P | | SNV | 9 |
| PP-3700 | 0 | p.L483P | | | |
| PP-57787 | 0 | p.L483P | | | |
| PP-59343 | 0 | p.L483P | | | |
| PP-59926 | 0 | p.L483P | | | |
| PP-60060 | 0 | p.L483P | | | |
| Brain-S5 | 0 | p.L483P | | | |
| PP-41342* | 0 | p.L483P/ p.E365K | | | |
| PP-3429 | 0 | p.A495P | | | |
| PP-3762*,PP-42378*,PP-3476,PP-3179,PP-3001,PP-3173,PP-3023,PP-42444,PP-3406,PP-56534,PP-52772,PP-41705 | 0 | No GBA variants | | | 12 |

Samples with * were discordant with BWA-GATK.
[a]Samples with IDs starting with NA- and HG- were obtained from NHGRI; samples with IDs starting with PP were obtained from PPMI; samples marked as brain were obtained from QSBB.

**Table 2 Non-pathogenic CNVs in 1kGP and AMP-PD cohorts.**

| | 1kGP | | | PD | | | | | | LBD | |
|---|---|---|---|---|---|---|---|---|---|---|---|
| | European | African | Other | European | | African | | Other or unknown | | European | |
| | Control | Control | Control | Case | Control | Case | Control | Case | Control | Case | Control |
| CNL | 2 | 3 | 9 | 8[a] | 7 | 0 | 0 | 0 | 0 | 13[c] | 13 |
| CNG | 1 | 74 | 18 | 11[b] | 6 | 1 | 2 | 0 | 1 | 21[d] | 11[e] |
| Total | 503 | 661 | 1340 | 2227 | 1213 | 22 | 27 | 76 | 15 | 2598 | 1941 |

[a]Three out of the eight PD cases with non-pathogenic CN losses also have a pathogenic GBA variant (two samples have p.L483P and one sample has p.N409S).
[b]Four out of the 11 PD cases with CN gains also have a pathogenic GBA variant (three samples have p.L483P and one sample has p.N409S).
[c]One out of the 13 LBD cases with non-pathogenic CN losses also has a pathogenic GBA variant, p.L483P.
[d]Five out of the 21 LBD cases with CN gains also have a pathogenic or PD-related GBA variant (p.L483P, p.D448H, c.1263del+RecTL, p.T408M, and compound heterozygote p.L483P/p.D448H).
[e]One out of 11 LBD controls with CN gains also has a PD-related GBA variant, p.T408M.

the 1kGP dataset. In eight trios, the proband carried a GBA missense variant, one in the exons 9–11 homology region and 7 outside of this region. In all cases, the variants were inherited by one parent and no de novo variants were detected (Supplementary Table 6).

**Prevalence of GBA recombinant and non-recombinant variants in healthy, PD and LBD populations.** Gauchian allowed us to provide a comprehensive large-scale analysis of all classes of GBA mutations in healthy, PD and LBD populations. Non-pathogenic CNVs, where the breakpoints do not alter the GBA coding region,

**Table 3 GBAP1-like variants in the exons 9–11 homology region in 1kGP, PD and LBD cohorts.**

| | | p.A495P | p.L483P | p.D448H | c.1263del | RecNcil | | c.1263del+RecTL | | Total |
|---|---|---|---|---|---|---|---|---|---|---|
| | | | | | | CNL | Conversion | CNL | Conversion | |
| 1kGP | N = 2504 | 1 | 5 | 0 | 2 | 1 | 0 | 0 | 2 | 11 |
| PD | Case (N = 2325) | 3 | 14 | 1 | 0 | 1 | 2 | 1 | 0 | 22 |
| | Control (N = 1255) | 0 | 6 | 1 | 0 | 0 | 0 | 0 | 0 | 7 |
| LBD | Case (N = 2598) | 4 | 23 | 4 | 6 | 10 | 3 | 2 | 2 | 54* |
| | Control (N = 1941) | 2 | 0 | 1 | 0 | 0 | 0 | 0 | 0 | 3 |
| PD + LBD called by Gauchian | | 9 | 43 | 7 | 6 | 11 | 5 | 3 | 2 | 86 |
| PD + LBD called by BWA-GATK | | 9 (+11 FP) | 27 | 7 (+2 FP) | 0 | 0 | 1 | 0 | 0 | 44 |

*One sample is compound heterozygous for p.L483P and p.D448H.

were ten times more frequent in 'Africans' than 'Europeans' (11.3 vs 1.1%, Table 2). This was primarily driven by a striking difference in the prevalence of CNGs (10.8 vs. 0.6% for controls from both cohorts; p < 2.2e-16). Additionally, 'Africans' also had more copies gained, with a median gain of three copies compared to one for 'Europeans'.

As non-pathogenic CNVs in GBA have not previously been considered as possible PD or LBD risk factors, we compared these across the combined disease cohorts to their controls. We detected no difference in the prevalence of all non-pathogenic CNVs (1.10 vs 1.25%), CNGs (0.67 vs 0.63%) or CNLs (0.43 vs 0.63%, Table 2). Addressing SNVs next, we noticed that p.N409S was found at a very high frequency in the PD cohort of AMP-PD (in both cases and controls, 5.5 and 12.6% respectively), because of the recruitment of a large number of individuals with Ashkenazi Jewish ancestry[19], where it is very common. After excluding individuals carrying it from both cohorts for consistency, GBA variants were more common in each disease cohort than in the respective controls as expected (Table 4) (PD 7.8 vs 3.9%; LBD 11.7 vs 3.5%). This was also true for severe GBA variants[20] (PD 1.7 vs 0.8%; LBD 3.1 vs 0.1%). The overall OR for mutations in each disease against its controls was higher in LBD than in PD (3.68 v 2.07; p = 0.0098), and this was even more striking for the severe mutations (30.83 v 2.12; p = 0.0009).

We also noted that some individuals carried both a CNG and another GBA variant, mostly a GBAP1-like variant in the exon 9–11 homology region (Table 2), as also seen in the cohort analysed by ONT. There were no individuals with a CNG and another GBA variant in the 1kGP cohort.

Considering all samples analysed by Gauchian, seven out of 146 with a CNG also had a GBAP1-like variant, against 71 of 10,407 without a CNG (4.8 vs 0.68%; p value = 9.77e-5). Three additional individuals carried a non-GBAP1-like variant and a CNG. In the PD and LBD cohorts and their controls, nine of ten individuals carrying a CNG as well as a coding variant in GBA were patients (four PD and five LBD). One healthy control carried a CNG and p.T408M, which is a mild PD risk allele but does not cause GD. As we did not find any healthy controls with a CNG and a pathogenic variant, we considered whether the combination of both is more detrimental than a coding variant alone (excluding again p.N409S carriers, one of whom also had a CNG). We did not detect a significant added risk for disease in the combined PD and LBD AMP-PD data against their controls (OR for CNG and other variants vs other variants alone 2.31, 95% CI 0.37–45.01).

## Discussion
The recent dramatic improvements in sequencing techniques have allowed a much better understanding of human genetic variation, but several regions, including some key disease-related genes, have remained challenging. One example is GBA, responsible for the autosomal recessive lysosomal storage disorder GD[1], and one of the most important genetic determinants of risk for PD and the closely related LBD[21]. Here we present and validate Gauchian, a novel GBA caller for Illumina WGS data, capable of detecting SVs and SNVs within GBA. Using ONT targeted sequencing, we demonstrate that in the cases of discrepant calls between Gauchian and BWA-GATK analysis, the Gauchian calls are correct. We also demonstrate that a refined ONT amplicon-sequencing pipeline can detect reciprocal recombinants, and indels as well as mutations within homopolymers in coding exons. Importantly, both methods detect CNGs and CNLs arising from reciprocal recombination and allow straightforward classification of CNLs into those that do and do not affect the coding region, previously a complex task[8,22]. We thus provide two complementary new tools for fully resolving the GBA gene, which will be helpful to the community. Illumina WGS data can now be analysed robustly, and ONT targeted sequencing can be applied in a cost-effective way where analysis of GBA is sufficient (see cost details in Supplementary Table 7). Furthermore, the principles used in Gauchian can be applied to the analysis of other regions camouflaged due to gene duplication[11].

To explore the potential of Gauchian in the population and in disease contexts, we applied it to a total of 10,623 samples from the 1000 Genomes Project and PD and LBD cohorts with their controls from the AMP-PD initiative. This allowed us to provide the first large-scale data on CNVs, and to evaluate the frequency of all classes of GBA variants in PD and LBD with greater accuracy than before. Reciprocal recombinants in particular are likely missed in PD studies[16], including a recent targeted short-read study which detected none in 3402 patients[8], although one study using exome data with qPCR validation reported CNGs in 1.2% of PD and 0.7% of controls[22]. In non-diseased individuals, we noted that CNGs were more common in those with "African" ancestry, with greater copy number variability. These results are consistent with the greater African genetic diversity, with recent evidence of "African" genomes demonstrating unexplored structural variation[23] and more variability in copy numbers of SMN1, SMN2 and CYP2D6[14,15]. They further highlight the need to study non-European genomes, which has yielded additional insights into Alzheimer's disease[24] and is being expanded in PD[25].

In the PD and LBD cohorts, Gauchian analysis almost doubled the pathogenic variants detected in the homology region compared to BWA-GATK (86 vs 44) and eliminated false positives. We also performed a direct comparison of GBA variant frequencies between PD and LBD, after excluding the common p.N409S variant due to selection bias in the PD cohort[19]. The prevalence of GBA pathogenic or PD-risk variants was significantly higher in LBD than in PD (11.7 vs 7.8%), and this

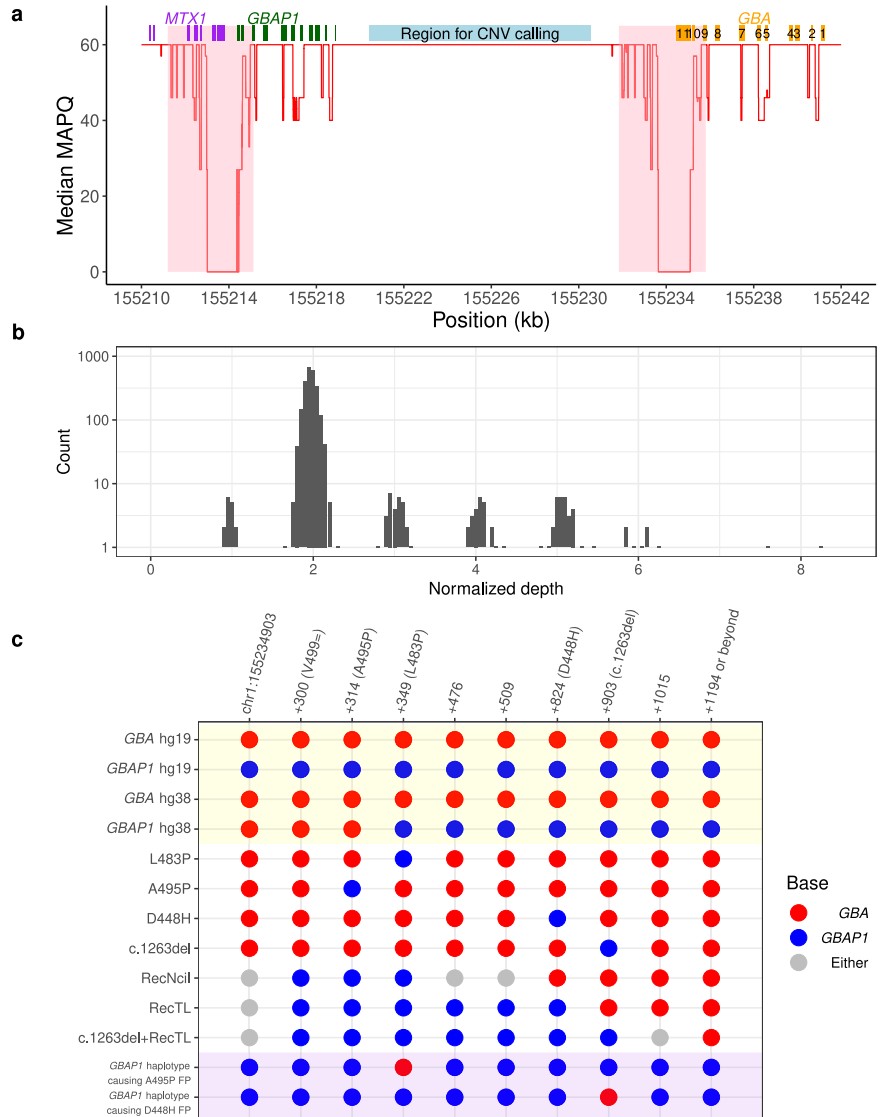

**Fig. 2 Gauchian detects challenging *GBA* variants through targeted copy number calling and haplotype phasing. a** Median mapping quality (red line) across 2504 1kGP samples plotted for each position in the *GBA/GBAP1* region (hg38). A median filter is applied in a 50 bp window. The eleven exons of *GBA* are shown as orange boxes. *GBAP1* and *MTX1* exons are shown as green and purple boxes, respectively. The 4 kb major homology region (98.1% sequence similarity, exons 9–11) between *GBA* and *GBAP1* is shaded in pink and highlights an area of low mapping accuracy. The light blue box shows the 10 kb unique region between the two genes in which copy number calling is performed in Gauchian. **b** Distribution of normalised depth in the 10 kb CN calling region in 2504 1kGP samples, showing peaks at CN1 (CNL), 2 (no CNV) and 3-8 (CNG). **c** Recombinant haplotypes in the exons 9–11 homology region, distinguished by *GBA/GBAP1* differentiating bases (x-axis). Reference genome sequences are shaded in yellow. There is an error in hg38 where the first three sites of *GBAP1* show *GBA* bases, which could lead to alignment errors. The *GBA* recombinant haplotypes are shown in the white background, including those where one or a few nearby sites are mutated to the corresponding *GBAP1* base, resulting from either gene conversion or CNL. Grey bases indicate that the base can be either *GBA* or *GBAP1* depending on the breakpoint position of the CNL/conversion. Shaded in purple are two example *GBAP1* haplotypes, found by Gauchian, that have been partially converted to *GBA* and can cause false-positive *GBA* variant calls by standard secondary analysis pipelines. For the first example, the reverse-p.L483P variant on *GBAP1* directs aligners to align *GBAP1* reads to *GBA*, causing the nearby p.A495P false-positive call. For the second example, the reverse-c.1263del variant inserts 55 bp to *GBAP1*, driving *GBAP1* reads to align to *GBA*, causing the nearby p.D448H false-positive call. CNG copy number gain, CNL copy number loss, CNV copy number variant.

difference was even larger for severe pathogenic variants (3.1 vs 1.7%). The OR for *GBA* mutations in LBD compared to controls was higher in our analysis than in the original report in this cohort[21] (3.68 v 2.90), and a previous study (2.55)[4], due to the detection of additional mutations and the filtering of p.N409S. *GBA* mutations increase the risk of cognitive decline in PD[26], and the odds ratio for *GBA* variants is higher in LBD than in PD with dementia[27]. Severe *GBA* variants in particular, which cause the neuronopathic form of GD[1], have a higher risk of PD[28] and a faster cognitive decline in PD than mild variants[29]. If PD and

LBD are considered as a spectrum of phenotypes with variable cognitive involvement, our findings further suggest that *GBA* variants, especially severe ones, tend to predispose to a phenotype on the LBD end of the spectrum. The main limitation of this analysis is the use of LBD and PD cohorts recruited separately, with the selection process necessitating the exclusion of p.N409S, and further comparisons in unselected matched cohorts are needed. The variable penetrance and phenotypic heterogeneity of PD and LBD patients with *GBA* mutations is attracting a lot of attention, with lysosomal gene variants acting as genetic

**Table 4 Summary of samples carrying *GBA* coding variants detected in 1kGP, PD and LBD cohorts.**

|  |  | p.N409S | Severe* variants | Total | Total excluding p.N409S |
|---|---|---|---|---|---|
| 1kGP | N = 2504 | 3 | 14 | 53 | 50 |
| PD | case N = 2325 | 128 | 38 | 296 | 171 |
|  | control N = 1255 | 158 | 9 | 200 | 43 |
|  | OR (95% CI) | n/a | 2.12 (1.07–4.71) | n/a | 2.07 (1.48–2.95) |
| LBD | case N = 2598 | 59 | 79 | 353 | 298 |
|  | control N = 1941 | 19 | 2 | 86 | 67 |
|  | OR (95% CI) | n/a | 30.83 (9.71–187.55) | n/a | 3.68 (2.82–4.87) |

*Severe and mild variants are defined in Supplementary Table 5.

modifiers[30]. An effect of common intronic haplotypes was suggested[31], but not seen by us in the AMP-PD cohort and part of the RAPSODI cohort used here[32]. A possible influence of CNGs has not yet been investigated. Although these do not alter the *GBA* coding region, they could affect expression and function, for example by acting as a competing endogenous microRNA sponge[33]. CNGs were not enriched in PD or LBD. There were, however, rare carriers of both a CNG and a pathogenic or PD-associated *GBA* variant. In the PD and LBD cohorts, nine out of ten of these were patients and the only control coding variant was the mild PD risk allele p.T408M. This raises the possibility that CNGs are modifiers, increasing the penetrance of other *GBA* variants. We have not, however, phased the CNGs and other variants, and did not show a statistically significant increased risk for carriers of a CNG and mutation compared to mutation alone. Therefore, further population and mechanistic work is required.

In conclusion, we have demonstrated that SNV detection and complete resolution of all classes of SVs is possible using the novel Gauchian caller with Illumina WGS, which outperforms BWA-GATK analysis, or with targeted ONT sequencing. We also demonstrate that CNVs are relatively common, and suggest that these merit investigation as possible modifiers of PD or LBD risk. Given the importance of this gene and the rapid progress to targeted clinical trials in PD[34], we propose that the adoption of either workflow should be considered by research and diagnostic labs, based on local resources and data availability.

## Methods

**Population cohorts and samples used**. We downloaded WGS CRAM files from the 1kGP. These were generated by 2 × 150 bp reads on Illumina NovaSeq 6000 instruments from PCR-free libraries sequenced to an average depth of at least 30x and aligned to the human reference, hs38DH, using BWA-MEM v0.7.15. We downloaded WGS CRAM files from PD[19] and DLB[21] cohorts and their controls from the AMP-PD knowledge portal. These were generated by sequencing 2 × 150 bp reads to >25x coverage and processing against hs38DH using the Broad Institute's implementation of the Functional Equivalence Pipeline[35]. We also downloaded AMP-PD variant calls, generated using BWA-GATK. Where samples had been recorded as European/Caucasian/white, or African/black in the original database, we refer to them as 'European' or 'African' for consistency and simplicity, despite the lack of scientific validity, in quotation marks as suggested[36].

Selected DNA samples were obtained from the PPMI[37] and the NHGRI Sample Repository for Human Genetic Research at the Coriell Institute for Medical Research. DNA samples from living individuals for ONT analysis were obtained from a clinical cohort, RAPSODI[38], which aims to define the risk of PD in *GBA* mutation carriers. Recruitment and analysis are ongoing, and the clinical results will be reported separately. DNA from saliva (Oragene DNA OG-500 kit, DNA Genotek) was extracted according to the manufacturer's protocol. Brain samples from 16 PD patients were obtained from the Queen Square Brain Bank, and DNA was extracted with phenol-chloroform[39] or MagAttract HMW DNA kit (Qiagen) from the frontal cortex, cerebellum or midbrain. Ethics approval was provided by the National Research Ethics Service London—Hampstead Ethics Committee for RAPSODI, NRES Committee central—London for QSBB samples, and UCL Ethics Committee for PPMI samples. All participants provided informed consent.

**Gauchian— a WGS-based *GBA* caller**. Gauchian builds upon the strategies to solve closely related paralogs, as described in our previously developed *SMN1/2* and *CYP2D6* callers[14,15]. Gauchian calculates the total number of copies of *GBA*,

*GBAP1* and *GBA/GBAP1* gene hybrids. Reciprocal recombinations across homologous regions lead to CNG and CNL of the 20.6 kb region between the homologous parts of the two genes. Since the breakpoint may vary in position, to detect CNVs, Gauchian uses the sequencing depth in the 10 kb unique region between *GBA* and *GBAP1* (chr1:155220429-155230539; hg38) (Fig. 2a). The number of reads aligned to this region is normalised and corrected for GC content, and the copy number is called from a Gaussian mixture model (Fig. 2b). A deviation of this copy number (CN) from the diploid expectation indicates the presence of a CNV, e.g. one copy indicates a CNL, and three or more copies indicate a CNG. Thus, this number plus two gives the total copies of *GBA* and *GBAP1* combined, i.e. CN (*GBA* + *GBAP1*). Included in this CN calculation, in addition to *GBA* and *GBAP1* genes, are gene hybrids where part of *GBA* and *GBAP1* are fused. CNG always leaves an intact copy of *GBA*, while CNL can create pathogenic *GBA-GBAP1* fusions if the deletion breakpoint falls within the *GBA* gene coding region.

Next, Gauchian identifies the breakpoint of the CNV, following a similar approach as previously described[15]. To do this, we identified 82 reliable sites (Supplementary Table 8) that differ between *GBA* and *GBAP1*. Gauchian estimates the *GBA* CN at each *GBA/GBAP1* differentiating site based on CN (*GBA* + *GBAP1*) and the numbers of reads supporting *GBA*- and *GBAP1*-specific bases. CNV breakpoints are identified when the CN of *GBA* changes. For example, a transition between CN1 and CN 2 indicates the breakpoint of a CNL, and a transition between CN 3 and CN 2 indicates the breakpoint of a CNG. The exact breakpoint is further refined by haplotype phasing as described in the next paragraph.

To identify recombinant variants, Gauchian analyses the 1.1 kb homology region in exons 9–11 (Fig. 2c) containing ten *GBA/GBAP1* differentiating sites that are 14–315 bp away from each other, several of which are critical *GBAP1-like* variants. These include p.L483P, p.D448H, c.1263del55, RecNciI, RecTL and c.1263del+RecTL (Fig. 2c). The high homology and the frequent gene conversion between *GBA* and *GBAP1* make exons 9–11 a challenging region for standard secondary analysis pipelines, which often miscall variants due to misalignments of recombinant variant reads. Additionally, three positions in the *GBAP1* reference sequence in hg38 erroneously contain the *GBA* bases (Fig. 2c, yellow shading), so *GBA* p.L483P reads would likely align to *GBAP1*, causing false-negative calls (*GBA/GBAP1* is among the regions enriched for discordant variant calls between hg19/hg38[40]). In addition, we found *GBAP1* haplotypes that have been partially converted to *GBA*. Those converted bases would direct *GBAP1* reads to align to *GBA*, causing false-positive *GBA* variant calls at nearby positions (Fig. 2c, purple shading). Gauchian takes a novel approach that does not rely on accurate alignments between *GBA* and *GBAP1*. Based on the linking information of reads and read pairs covering the ten differentiating sites in either *GBA* or *GBAP1*, Gauchian phases all the haplotypes at these sites originating from either *GBA* or *GBAP1* and identifies hybrid haplotypes (i.e. a mixture of *GBA* and *GBAP1* bases on the same haplotype). This allows us to identify CNL breakpoints as well as small and big gene conversion events.

To assess the relative abundance of the different haplotypes, Gauchian uses CN (*GBA* + *GBAP1*) and haplotype-supporting read counts at the differentiating bases to call the CN of each haplotype. Gauchian compares two scenarios: one copy of the wild-type *GBA* haplotype vs. two copies of the wild-type *GBA* haplotype. Gauchian determines which scenario is more likely given the number of supporting reads in the data. If we call only one copy of the wild-type *GBA* haplotype, this indicates that the individual is a carrier of the disease-causing variant. If an individual is a carrier of more than one variant haplotype and there is no haplotype that carries the *GBA* base at all variant sites of interest, Gauchian calls this sample compound heterozygous. Homozygous variants are called when the CN of the *GBA* base is called 0.

In addition to *GBAP1*-like variants in exons 9–11 homology region, Gauchian targets all known *GBA* pathogenic or likely pathogenic variants as classified by ClinVar (Supplementary Table 9), including non-*GBAP1*-like variants, and *GBAP1*-like variants outside the exons 9–11 homology region. For these, since variants don't correspond to *GBAP1*, or, if they do, the region between *GBA* and *GBAP1* is not highly similar and alignments are accurate, Gauchian parses read alignments and calls the CN of variants based on the number of variants supporting reads as described for *SMN/CYP2D6* callers. Gauchian is a targeted

caller for known variants and thus does not call novel variants. The analysis takes roughly 1 min per sample when run on a standard machine using a single thread and can be speeded up with multiprocessing.

**ONT long-read sequencing with PCR enrichment**. *GBA* enrichment was obtained via PCR (Supplementary Table 10), with primer pair 1, as previously described[41], modified to carry the ONT barcode adaptor sequence. The product was an 8.9 kb amplicon containing the entire *GBA* coding region and introns (chr1:155232524-155241392; hg38). Samples were barcoded using the 96-sample barcoding kit (EXP- PBC096). Amplicons were purified with Agencourt AMPure XP magnetic beads at a ratio of 0.4x. Library preparation was carried out according to the ONT protocol (version: PBAC96_9069_v109_revO_14Aug2019 – long fragment selection) and sequencing with a MinION device on R9.4 flow cells.

Primary acquisition of sequencing data was carried out with MinION (version 20.10.3)[42], and base-calling and demultiplexing with Guppy (version 4.2.2). The resulting reads were aligned to GRCh38.p13, without the alternative reference contigs, using NGMLR (version 0.2.7)[43] unless otherwise stated. Clair (version 2.1.1)[44] was used for SNV calling. Since the most recent Clair ONT models were trained with up to 578-fold coverage, each sample was down-sampled to 550-fold. SNV calls were filtered with the Clair genome quality (GQ) score with a threshold set at 650. We only called SNV in *GBA* coding exons and ten flanking bases. Intronic haplotypes for some of these samples with no coding mutations were recently reported separately[32]. Phasing of SNV was carried out with Whatshap (version 1.0)[45] and data manipulation with Samtools (version 1.10)[46], Bedtools (version 2.29.1)[47] and Tabix (version 1.7-2). Optimisation from our previous method[16] comprised Guppy instead of Albacore for base-calling and Clair instead of Nanopolish for SNV calling. The pipeline used here identified all previously reported coding SNVs in 95 samples which were re-analysed. Analysis was run on a machine with intel 4 cores/8 threads, 16 Gb RAM DDR4 and 1 Tb SSD memory drive. Guppy was run on the UCL computational cloud, using 2 NVIDIA A100 GPUs in parallel. The total analysis took 2 h for a maximum of 96 samples.

To detect homopolymers, we devised a method that involves analysis of.bam files with S*amtools depth* to obtain the depth of coverage across these *A* (chr1:155239990-155239995 and chr1:155239657-155239661; hg38). The depth of coverage at each position was then adjusted for the depth of coverage at the 100 flanking positions, and the result was compared with the mean of all other samples in the run. If the adjusted depth of coverage at one position was more than five median absolute deviations from the median adjusted depth of coverage of the other samples in the run at that position, this was considered evidence of a deletion (coverage lower than the mean for that position) or an SNV (coverage higher than the mean for that position). Variants detected with this method were validated with Sanger sequencing[48].

To detect and amplify reciprocal recombination events, two additional sets of primers were used[41]. These primers were specifically designed to amplify the recombinant alleles: the set *MTX1-r/GBA-nf* (primer pair 2) only amplifies recombinants with *GBA* gene sequence at the 5′UTR end and *GBAP1* sequence at the 3′-UTR end (CNL), while the set *ΨMTX1-r/ΨGBA-nf* (primer pair 3) only amplifies recombinants with the 3′-UTR end and *GBAP1* sequence at the 5′-UTR end (CNG, see Fig. 1). Samples underwent PCR using these pairs of primers. If a product was detected on agarose electrophoresis, the sample was re-amplified with the same primer pair modified to carry the ONT barcode adaptors, and the amplicons were barcoded and sequenced as described. The primer sequences and PCR conditions are given in Supplementary Table 11. To define the breakpoint of CNL, the products of primer pair 2 were aligned to *GBA* (to avoid alignment to *GBAP1;* chr1:155222384-155241249; hg38) using LAST (Version 1243)[49]. The resulting alignment was analysed with Clair to look for variants at positions where *GBA* and *GBAP1* differ (Supplementary Table 8). If a sample displayed an SNV in a certain position, it meant that the breakpoint must be upstream of that but downstream of the next sentinel position where no variant is detected (Supplementary Fig. 3).

**GBA enrichment with UNCALLED**. To validate *GBA* SV without PCR enrichment, we used UNCALLED, which uses adaptive sampling to enable real-time enrichment or depletion on MinION runs via the MinKNOW API ReadUntil[17]. UNCALLED analyses the signal generated by the DNA molecules passing through each pore of the device in real-time and decides whether they align to a reference sequence provided. It can then prematurely eject the molecule from the pore if not of interest, freeing up sequencing capacity for new reads and ultimately achieving purely computational enrichment. The target region for enrichment was chr1:155193567-155264811, with repetitive regions masked with the UCSC RepeatMasker.

**Digital PCR**. dPCR was performed by QIAGEN (Hilden, Germany) on the QIA-cuity instrument. Three probes were selected: DCH101-0776005A (chr1:155231010-155231209; hg38) and DCH101-0776012A (chr1:155232410-155232609; hg38) target the region affected by the recombination event, while DCH101-1260927A (chr1:155208699-155208804; hg38) is outside of this region and was used as a reference for the analysis. Each sample was tested three times, and the result is the average of the three assays.

**Illumina sequencing**. WGS was performed on the Illumina NovaSeq instruments using Illumina TruSeq Nano DNA Library Prep[9].

**Statistics and reproducibility**. Analysis was carried out on R (version 4.0.5). Odds Ratios (OR) were calculated with logistic regression. To check for an additive effect on the risk of CNGs on *GBA* variant carriers, multivariate logistic regression was used, with disease status as the outcome variable, and *GBA*-carrier status and CNG-carrier status as independent variables. The sample size is reported in Tables 2, 3.

**Reporting summary**. Further information on research design is available in the Nature Research Reporting Summary linked to this article.

## Data availability

Gauchian can be downloaded from: https://github.com/Illumina/Gauchian ONT and UNCALLED scripts used can be downloaded at https://github.com/marcotoffoli Individual-level genome sequence data for the PD patients, LBD patients, and neurologically healthy controls are available at AMP-PD (https://amp-pd.org). The datasets of DNA from QSBB brain samples and NHGRI samples generated during this study (Illumina WGS and targeted ONT sequencing) will be made available on the European Nucleotide Archive (https://www.ebi.ac.uk/ena/browser/home), accession number PRJEB48317. The datasets only include read alignments to *GBA/GBAP1* regions (other regions of the genome have been removed or masked) to minimise the amount of genetic information made available for public access. The datasets of DNA from PPMI samples generated during this study (targeted ONT sequencing) are incorporated into the aggregated 'Current Biospecimen Analysis Results' dataset in the PPMI repository. The data were clearly labelled as project 195 data (https://www.ppmi-info.org/). ONT sequencing data on living individuals are not available due to consent/IRB restrictions. 1kGP project https://www.ncbi.nlm.nih.gov/bioproject/PRJEB31736 All other data are available from the corresponding author on reasonable request, when compatible with consent/IRB restrictions. RAPSODI, https://rapsodistudy.comOMIM, http://www.omim.org/ The NCBI reference sequence for *GBA* on which the numbering of exons is based is NM_000157.4.

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

## Acknowledgements

This study was supported in part by the Intramural Research Programme of the National Institutes of Health (National Institute of Neurological Disorders and Stroke; project numbers: 1ZIANS003154) and the JPND through the MRC grant code MR/T046007/1.

We thank the New York Genome Center and the Coriell Institute for Medical Research for generating and releasing the 1kGP WGS data. We thank the AMP-PD Knowledge Platform for hosting WGS data for patient and control cohorts.

We thank Salomé Bagayan, Diem Cao, Angela Henry, Wen Liang, Leland Mencik, Lisa Robison, Christina Toma and Sasha Treadup at Illumina for helping with the sequencing of validation samples, and Iska Steffens at QIAGEN for performing digital PCR.

Data and biospecimens used in the analyses presented in this article were obtained from the Parkinson's Progression Markers Initiative (PPMI) (www.ppmi-info.org/specimens). As such, the investigators within Revised April 2019 PPMI contributed to the design and implementation of PPMI and/or provided data and collected biospecimens, but did not participate in the analysis or writing of this report. For up-to-date information on the study, visit www.ppmi-info.org.

PPMI—a public-private partnership—is funded by The Michael J. Fox Foundation for Parkinson's Research and funding partners, including [list the full names of all PPMI funding partners found at www.ppmi-info.org/fundingpartners].

Data used in the preparation of this article were obtained from the AMP-PD Knowledge Platform. For up-to-date information on the study, https://www.amp-pd.org.

AMP-PD—a public-private partnership—is managed by the FNIH and funded by Celgene, GSK, the Michael J. Fox Foundation for Parkinson's Research, the National Institute of Neurological Disorders and Stroke, Pfizer, Sanofi and Verily.

Clinical data and biosamples used in preparation of this article were obtained from the Fox Investigation for New Discovery of Biomarkers (BioFIND), the Harvard Biomarker Study (HBS), the Parkinson's Progression Markers Initiative (PPMI) and the Parkinson's Disease Biomarkers Programme (PDBP).

BioFIND is sponsored by The Michael J. Fox Foundation for Parkinson's Research (MJFF) with support from the National Institute for Neurological Disorders and Stroke (NINDS). The BioFIND Investigators have not participated in reviewing the data analysis or content of the manuscript. For up-to-date information on the study, visit michaelj-fox.org/biofind.

Harvard NeuroDiscovery Biomarker Study (HBS) is a collaboration of HBS investigators [full list of HBS investigator found at https://www.bwhparkinsoncenter.org/biobank] and funded through philanthropy and NIH and Non-NIH funding sources. The HBS Investigators have not participated in reviewing the data analysis or content of the manuscript.

Parkinson's Disease Biomarker Programme (PDBP) consortium is supported by the National Institute of Neurological Disorders and Stroke (NINDS) at the National Institutes of Health. A full list of PDBP investigators can be found at https://pdbp.ninds.nih.gov/policy. The PDBP Investigators have not participated in reviewing the data analysis or content of the manuscript.

We are grateful to the Queen Square Brain Bank, and to individuals who donated their brains. The Queen Square Brain Bank is supported by the Reta Lila Weston Institute for Neurological Studies and the Medical Research Council UK.

## Author contributions

MT Acquisition, analysis and interpretation of data. Prepared first manuscript draft. Creation of software to detect changes in homopolymers in ONT data. XC Acquisition, analysis and interpretation of data. Prepared first manuscript draft. Creation of Gauchian software. FJS Conception, design and supervision of the work. Substantively revised the manuscript. C-YL Acquisition of data. SM Substantive contribution to conception and design of the work. Revised the manuscript. AH Acquisition of data. SK Acquisition of data. MEG-S Acquisition of data. ES Acquisition of data. SWS Substantive contribution to conception and design of the work. Revised the manuscript. AHVS Conception, design and supervision of the work. Substantively revised the manuscript. MAE Conception, design and supervision of the work. Substantively revised the manuscript. CP Conception, design and supervision of the work. Substantively revised the manuscript. All authors read and approved the final manuscript.

## Competing interests

XC and MAE were employees of Illumina Inc. during the research and writing of this manuscript and are now employees of Pacific Biosciences Inc. SWS serve on the Scientific Advisory Council of the Lewy Body Dementia Association. SWS is an editorial board member for the Journal of Parkinson's Disease and JAMA Neurology. AHVS is supported by the UCLH NIHR BRC. MT, FJS, C-YL, SM, AH, SK, MEG-S, ES and CP declare no competing interests.

## Ethics

Ethics approval was provided by the National Research Ethics Service London—Hampstead Ethics Committee for RAPSODI, NRES Committee central—London for QSBB samples and UCL Ethics Committee for PPMI samples. All participants provided informed consent.
