## [Peer Review File · Communications Biology]

Referee expertise:

Referee #1: computational genomics, CNVs

Referee #2: bioinformatics, -omics, genetics of common and rare diseases

Reviewers' comments:

Reviewer #1 (Remarks to the Author):

The Toffoli et al. manuscript presents two methods for genotyping variants in the GBA gene. GBA is known to carry alleles causing the recessive Gaucher disease and increasing the risk for Parkinson's disease (PD) and the related Lewy body dementia (LBD), justifying the efforts to study the GBA alleles that segregate in the population. Unfortunately, the GBA locus is complicated due to the nearby pseudogene GBAP1 that has large sequence similarity with GBA obfuscating genotyping through short reads. The authors address this complication by developing a method they call Gauchian that manages to make sense of short reads obtained by whole-genome sequencing by taking advantage of the sequence structure of the GBA-GBP1 locus. In addition, the authors present a second method that uses PCR enrichment and long-read Nanopore sequencing to genotype the complicated locus.

The authors show that Gauchian and the long-read method agree when both methods are applied to the same sample. Moreover, they show that Gauchian generates GBA genotypes of higher quality (more variants detected while decreasing the number of false-positive calls) than the standard GATK pipeline designed to operate on the whole genome. Alleles that have already been reported in GBA are readily genotyped, including SNVs and more structural rearrangements.

Gauchian is directly applicable to available whole-genome sequencing data from controls or PD and LBD cases, and the authors proceeded to genotype ~10,000 samples. The results provide an extensive and improved population view of the allele-spectrum in the GBA gene, demonstrating that 'Africans' carry substantially more of the copy number gains and that PD and LBD cases carry significantly more damaging alleles than controls.

It is worthwhile to develop specialized methods for the many complex regions across the human genome. The authors have a strong record in that direction, having published methods for SMN1, SMN2, and CYP2D6 loci and a prior effort on the GBA locus. The work is solid and generally well described, and I have hardly anything to suggest.

One area that seems like a missed opportunity is the attempt to quantify the rate of de novo variants in GBA regions. It seems easy to apply Gaussian over the available whole-genome sequencing for trios, and the locus should be fairly frequently mutated to maintain such a high number of different alleles. Having a rough estimate of the de novo rate will be helpful in getting a deeper understanding of the locus.

I have two minor issues:

- I find the title is cryptic and confusing. I would suggest something like 'Comprehensive analysis of structural re-arrangements in the GBA gene using ...'
- It would be helpful to describe the RecNciI and RecTL alleles before first use, probably in the introduction.

Reviewer #2 (Remarks to the Author):

The authors described a method called Gauchian to detect genetic variants (SNV, CNV and SV) in the GBA gene region complicated by pseudogene GBAP1, and validated the method using ONT technology. While the method seems promising in terms of accuracy according to analysis results presented in this manuscript, two major questions need to be answered

1. Have the authors tried to implement the short-read data based method to other regions with pseudogenes in the human genome, and if so what is the performance of the method?
2. What is the comparison of costs between these two strategies (Gaussian and ONT), both computationally and economically?

Point-by-point response to reviewers

We would like to thank the reviewers for taking the time to read our paper and for providing their helpful comments. We have addressed the points raised with amendments marked with changes in red in the revised version of the manuscript. We believe that with the suggested changes the overall quality of the manuscript has improved.

We have included a black version of the supplemental material and a version of the same file with changes in red.

We would also like to highlight that the authors XC and MAE changed affiliation since the first submission. We have amended the manuscript to reflect this.

Below we report the comments from each reviewer (in black) and our answers to the points raised (in red).

Reviewer #1 (Remarks to the Author):

The Toffoli et al. manuscript presents two methods for genotyping variants in the GBA gene. GBA is known to carry alleles causing the recessive Gaucher disease and increasing the risk for Parkinson's disease (PD) and the related Lewy body dementia (LBD), justifying the efforts to study the GBA alleles that segregate in the population. Unfortunately, the GBA locus is complicated due to the nearby pseudogene GBAP1 that has large sequence similarity with GBA obfuscating genotyping through short reads. The authors address this complication by developing a method they call Gauchian that manages to make sense of short reads obtained by whole-genome sequencing by taking advantage of the sequence structure of the GBA-GBP1 locus. In addition, the authors present a second method that uses PCR enrichment and long-read Nanopore sequencing to genotype the complicated locus.

The authors show that Gauchian and the long-read method agree when both methods are applied to the same sample. Moreover, they show that Gauchian generates GBA genotypes of higher quality (more variants detected while decreasing the number of false-positive calls) than the standard GATK pipeline designed to operate on the whole genome. Alleles that have already been reported in GBA are readily genotyped, including SNVs and more structural rearrangements.

Gauchian is directly applicable to available whole-genome sequencing data from controls or PD and LBD cases, and the authors proceeded to genotype ~10,000 samples. The results provide an extensive and improved population view of the allele-spectrum in the GBA gene, demonstrating that 'Africans' carry substantially more of the copy number gains and that PD and LBD cases carry significantly more damaging alleles than controls.

It is worthwhile to develop specialized methods for the many complex regions across the human genome. The authors have a strong record in that direction, having published methods for SMN1, SMN2, and CYP2D6 loci and a prior effort on the GBA locus. The work is solid and generally well described, and I have hardly anything to suggest.

1) One area that seems like a missed opportunity is the attempt to quantify the rate of de novo variants in GBA regions. It seems easy to apply Gaussian over the available whole-genome sequencing for trios, and the locus should be fairly frequently mutated to maintain such a high number of different alleles. Having a rough estimate of the de novo rate will be helpful in getting a deeper understanding of the locus.

This is a very valid point. We analysed trios in the 1kGP and found 8 cases where the proband carried a missense variant in *GBA*. All of these were inherited in a mendelian pathway, and no *de novo* variants were detected. We described this in the results section (lines 205-210) and in supplementary table 6.

I have two minor issues:

2) I find the title is cryptic and confusing. I would suggest something like ‘Comprehensive analysis of structural re-arrangements in the GBA gene using ...’

We have amended the title removing the words “a novel algorithm for” to make it easier to read. However, we can’t apply the reviewer’s suggestion in full, as the methods described detect not only structural re-arrangement, but the full range of variants within the GBA region.

3) It would be helpful to describe the RecNcil and RecTL alleles before first use, probably in the introduction.

We agree with the reviewer, the nomenclature for these alleles can be confusing.

We initially described these recombinant alleles in the methods section, but we moved this to the introduction at lines 89-96.

Reviewer #2 (Remarks to the Author):

The authors described a method called Gauchian to detect genetic variants (SNV, CNV and SV) in the GBA gene region complicated by pseudogene GBAP1, and validated the method using ONT technology. While the method seems promising in terms of accuracy according to analysis results presented in this manuscript, two major questions need to be answered

1. Have the authors tried to implement the short-read data based method to other regions with pseudogenes in the human genome, and if so what is the performance of the method?

We thank the reviewer for raising this point. Each of the regions affected by pseudogenes contain their own unique problems that need to be addressed individually. Gauchian is specifically designed for the *GBA-GBAP1* region and won’t apply to other regions directly. We have previously developed targeted callers for two other regions with high sequence homology problems and they achieved high accuracy: *SMN1/SMN2* [1] and *CYP2D6/CYP2D7* [2]. We have also mentioned this in the discussion “Furthermore, the principles used in Gauchian can be applied to analysis of other regions “camouflaged” due to gene duplication” (lines 289-290).

References:

1. Chen X, Sanchis-Juan A, French CE, Connell AJ, Delon I, Kingsbury Z, et al. Spinal muscular atrophy diagnosis and carrier screening from genome sequencing data. *Genet Med.* 2020;22:945–53.
2. Chen X, Shen F, Gonzaludo N, Malhotra A, Rogert C, Taft RJ, et al. Cyrius: accurate CYP2D6 genotyping using whole-genome sequencing data. *Pharmacogenomics J.* 2021;21:251–61.

2. What is the comparison of costs between these two strategies (Gauchian and ONT), both computationally and economically?

To provide insight into the costs and computational requirements for both techniques, we added the computational requirements in methods (lines 431-432 and 456-458), and ONT cost details in Supplementary table 7, which we refer to in the discussion (lines 288-289).

Gauchian is an add-on bioinformatic analysis to short read WGS, with no economic cost when WGS data is available.

REVIEWERS' COMMENTS:

Reviewer #2 (Remarks to the Author):

The response answers my questions. Thank you.